# The Role of Preventative Medicine Programs in Animal Welfare and Wellbeing in Zoological Institutions

**DOI:** 10.3390/ani13142299

**Published:** 2023-07-13

**Authors:** Paolo Martelli, Karthiyani Krishnasamy

**Affiliations:** 1Ocean Park Hong Kong, Hong Kong, China; 2Ocean Park Conservation Foundation, Hong Kong, China; karthimartelli@yahoo.com.sg

**Keywords:** preventative medicine, animal welfare, animal wellbeing, operant conditioning

## Abstract

**Simple Summary:**

In animals undergoing veterinary treatment, including therapeutic, quarantine and preventative medicine, their physiology and behaviour are derailed from their normal states and the animal’s choices and comfort are de facto limited. Traditional paradigms of animal welfare do not apply perfectly to animals under veterinary care, including quarantine and preventative medicine. A paradigm separating animal wellbeing from animal welfare is proposed in order to instil much needed clarity of thought and to guide actions in regard to the welfare of animals under human care. Using such a model, preventative medicine programs emerge as a cornerstone of zoo and aquarium animal welfare. Preventative medicine programs tailored to individual animals and organizations must weigh the expected benefits versus the risks inherent associated with veterinary procedures. As veterinary medicine and husbandry science continue to advance, preventative veterinary procedures can be carried out more safely and more frequently. The inclusion of operant conditioning to zoo veterinary practice allows the voluntary participation of the animals, thereby reducing the associated risks and costs. It also facilitates the collection of abundant and reliable physiological data, including the indicators of animal wellbeing, to help objectively evaluate the adequacy of the animal welfare.

**Abstract:**

The overarching goal of a preventative medicine program is to minimize the chances of health problems developing and to maximize the chances of detecting health problems early, in a manner that best benefits the animals and the organization. The traditional paradigms of animal welfare, stemming from the five freedoms and being progressively fleshed out to five domains, the 24/7 approach and so forth do not apply perfectly to zoological collections and less so to animals undergoing veterinary treatments. The physiology and behaviour of animals undergoing veterinary treatments, including therapeutic, quarantine and preventative medicine, are derailed from their normal states and their choices and comfort are de facto limited. A paradigm separating animal wellbeing from animal welfare is necessary to instil clarity of thought and to guide actions in regard to the welfare of animals under human care. Using such a model, preventative medicine programs emerge as a cornerstone of zoo and aquarium animal welfare, all the more if it incorporates modern veterinary and husbandry techniques, including operant conditioning.

## 1. Reframing the Paradigm to Differentiate Welfare from Wellbeing

The fundamentals of captive animal welfare find their origins in Brambell’s early work introducing the “five freedoms” [1]. This evolved further into the “Five Domains”, aimed at facilitating the systematic, comprehensive and coherent assessment of animal welfare [2,3,4,5]. Over time, models that included the length and complexity of animals’ lives and the role of the caretakers also contributed to give more substance to the topic [6,7,8]. With every attempt at refining the animal welfare models, the definition of welfare widens and the number of criteria used to assess it increases [5,9,10,11,12,13]. To provide adequate welfare under captive conditions many components must be considered, delivered and evaluated. As mentioned by Fraser [14,15], some of these components may be at odds and attempts to improve welfare in one area may diminish it in other areas. Regardless of which models and concepts are adopted, all unequivocally agree that good health is a necessary condition in order for the claim of good animal welfare to be made [14,15,16,17].

Disease processes adversely disrupt the mental and physiological state of the patient. To diagnose and treat diseases or to perform preventative medicine examinations, procedures that are physically and psychologically intrusive and may cause some degree of discomfort, including stress, fear, pain, temporary confinement, various drug effects and side effects, are often necessary. That is why caveats such as “veterinary discretion applies” or “excluding animals under veterinary treatment” are often included, and invariably implied, in documents pertinent to animal welfare, such as husbandry manuals, minimum dimension requirements, etc.

The widely accepted animal welfare framework outlined above, though useful, is not always relatable to animals in zoological collections and even less so to animals undergoing veterinary treatments. These models imply that the welfare of sick animals and even of animals undergoing preventative care is inevitably deficient. This view is not conducive to systemic improvements, and we must move past it to appreciate the importance of preventative medicine programs in animal welfare. Discussions on the welfare of zoo animals are complex and can often lead to intellectual or conversational impasses [12,14,15,18,19,20]. Perhaps this can be traced to the loose interchangeability and the implied equivalence, both in the literature and in the vernacular, of the terms (animal) “welfare” and (animal) “wellbeing” [4,5,9,10,13,15,16,21,22]. We propose to use distinct definitions of animal welfare and animal wellbeing as shown in Table 1.

The deep substantial differences between wellbeing and welfare are obvious when applied to humans, who have complex perceptions of self, time and space and live in highly organized societies. Human welfare refers to a system of governance, the welfare state, about which there are different models and viewpoints [24,25,26,27,28]. The purpose of human welfare is to govern (manage) to move society towards providing equal opportunities for all its members to reach a state of doing well. The intricate nature of human wellbeing represents a significant portion of the corpus of ancient and modern philosophy [28]. Human welfare measures are intended for the meekest, humblest, voiceless members of society.

Animals are unable to distinguish between the concepts of wellbeing and welfare and only experience various degrees of comfort, or a lack thereof. Williams recognises that human wellbeing is not the same as human welfare but proceeds to suggest extending the “componential view of human wellbeing to animal welfare” [29], thereby accepting the equivalence between the terms animal welfare and animal wellbeing. Other authors recognize the difference between welfare and wellbeing in humans and see a similarity between human wellbeing and animal welfare [16,30,31]. Franks et al. studying rats, accepts that the equivalent to human wellbeing in rats is animal welfare [32]. The definition of welfare of the World Organization for Animal Health (formerly Office International des Epizooties, OIE) does not distinguish between welfare and wellbeing either as it states that welfare is “how an animal copes with the conditions in which it lives” [32]. However, it also adds a list of management practices deemed necessary to help animals cope well with the living conditions [33]. This points to the unstated but implied suggestion that practices under exclusive human control regulate an animal’s wellbeing [13,33].

We suggest that by narrowing the definition of animal welfare rather than by expanding it, and by differentiating animal welfare from animal wellbeing and rejecting the equivalence of these terms, we can move past many obstacles. Following the human models, thus making a clear distinction between the concepts of welfare and wellbeing, adds clarity to the topic of animal welfare and animal wellbeing.

In the text hereafter, animal wellbeing refers to the current physical, mental and social health of an individual or group of animals. Wellbeing is a state internal to the animal and is not equivalent to animal welfare. The term animal welfare refers to the systems of management employed to achieve the maximum potential wellbeing of the individual or group of animals in zoological institutions. Consequently, animal welfare is external to the animals, is the responsibility of humans (under human care) and is not equivalent to animal wellbeing (Table 1).

Animal wellbeing is the observed output of animal welfare, and animal welfare comprises three categories of inputs. Two of these categories of inputs are largely outside human control yet determine much of the husbandry requirements and practices. The first category is the evolutionary history of the species, such as its social structures (gregarious, solitary, precocious, altricial, etc.), its natural diet, its natural habitat and habits (arboreal, aquatic, fossorial) and so forth. The second category is the animal’s individual history, such as age, sex, physical disabilities, temperament, past history, etc. Humans have no influence in the first category because it is the result of natural evolution that occurred even before our species existed. However, zoo managers must have the knowledge and resources to create captive environments suited to the animal’s innate needs. Humans only partly influence the second category, the individual history of the animal, because although it is an internal response by the animal, much of an individual’s history takes place while under human care. The third category of inputs consists of management practices. Management practices, although directed at the animals, are unquestionably and solely under human control and they are a reflection of current values and available resources. They are external to the animals who have no say in them.

The unambiguous distinction between animal wellbeing and animal welfare portrayed above can be illustrated with two examples.

In scenario one, imagine a poorly constructed chimpanzee enclosure where unsupervised visitors fling food and objects; there are no enrichment programs, staff are poorly trained and veterinary care is limited to the occasional emergency intervention. In this enclosure, imagine a young chimpanzee that gets along well with her cage mates and is presently in fine health. This particular animal enjoys a good wellbeing (internal state) even though its current welfare conditions are poor. The negligent management practices (poor welfare) are an impediment to its opportunities to prosper because they place the animal’s wellbeing at risk through a high likelihood of diseases, malnourishment and injuries and a low likelihood of these problems being recognized or corrected.

In the second scenario, a geriatric bear in an establishment with high standards of husbandry develops an untreatable neoplasia. This zoo has a comprehensive preventative medicine programme that allows the diagnosis of the bear’s disease before it is sensed by the bear. The bear’s health is in the hands of a proficient team of animal care professionals who expertly mitigate and assess the progression of the morbid processes. When the evolution of the disease eventually brings the bear near its humane endpoint, the bear is scheduled for euthanasia. Good management practices (welfare, external to the bear) identify and mitigate the disease processes (internal to the bear) in the best interest of the animal and of the organization. In this second scenario, while the disease obviously negatively affected the animal’s wellbeing, the bear’s welfare was indisputably of a high standard.

Animal welfare and animal wellbeing, as defined above, are closely linked. The quality of the animals’ welfare is an objective and fair measure of the abilities and of the commitment of the human carers to uphold the animals’ wellbeing.

This paradigm recognising the difference between wellbeing and welfare of the animal allows to further build on the constructive outlook presented by the “opportunities to thrive” models of Vicino and Miller [34]. The animals’ opportunities to experience their best possible wellbeing in a zoo increase with the quality of the animal welfare provided by that zoological institution [30].

## 2. The Role of Preventative Medicine Programs in Improving Animal Welfare Optimizing Animal Wellbeing

By adopting a paradigm that distinguishes between animal welfare and animal wellbeing we can discuss further and with more clarity the role of veterinary medicine in both the animal wellbeing and its welfare. In doing so, preventative medicine emerges as a pillar of zoo animal welfare.

The overall objective of a preventative medicine program is to minimize the chances of health problems developing by preventing their occurrence through the maximization of the chances of detecting health problems early or through averting devastating health outcomes [35] and mitigating complications resulting from the onset of disease [35,36].

Preventative medicine programs are relatively easy to cost, thereby aiding budgeting exercises, fundraising and allocation of funds. A comprehensive preventative medicine program allows for an efficient use of veterinary services, ensuring that there are resources to spare for ad hoc clinical treatments, staff training, exhibit upgrades, broader collection management and other matters that further contribute to animal welfare and to the animals’ wellbeing [8,16,17] in ways that best benefit the animals and the zoological institution. Preventative medicine addresses the needs of the animals and are tailored to the institution’s collection plan, exhibit designs, financial resources and staff expertise, while also weighing other influences, such as political, societal, ethical, cultural and religious values [37]. It is therefore not surprising that preventative medicine programs vary between institutions and are living documents that must be updated on a regular basis to incorporate current and future trends to advance the wellbeing of the collection.

Four broad positive outcomes are expected of a good preventative medicine program. Firstly, the prevention of avoidable diseases, through vaccination programs, regular deworming, balanced species-specific diets, harmonious social grouping and the monitoring of physiological data to detect predictors or prodromic signs of disease. Secondly, the early detection of morbid or degenerative processes that can be corrected to restore health. These include reversible organ systems impairment, dental pathologies, cataracts, benign neoplasia, nutritional imbalances, abnormal development issues, etc. Thirdly, as age advances or if premature diseases occur, preventative medicine programs are also tasked with designing and dispensing treatments to lessen the burden of morbid or degenerative processes that cannot be corrected fully. These include irreversible organ systems impairment, malignancies, age-related skeletal pathologies, etc. [35,36]. Lastly, euthanasia, practised to prevent prolonged suffering on the way to unavoidable death.

The financial, human and technical resources invested in order to maximise the outcomes must balance cost and the expected benefits relative to the risks [36,37].

Figure 1 shows a selection of preventative medicine procedures presented within a frame with four quadrants outlined by a vertical axis representing increasing cost or risk and a horizontal axis representing increasing potential benefits. Planned preventative measures can be positioned amongst the four quadrants. For obvious reasons there would be very few procedures with an expected low benefit regardless of risks or cost. Investigations that are able to yield a large benefit are the most desirable. Managerial efforts (animal welfare) devoted to reducing the risks and cost of veterinary procedures, such as new technologies, staff training and training animals for voluntary medical procedures (see more on this below), will improve the wellbeing of the animals [8,19].

Procedures planned as part of the preventative medicine program reflect the institutional commitment to and understanding of an animal’s needs. Planned procedures should detect reliable predictors or indicators of disease, leading to appropriate veterinary or management decisions, and they must be worth the inherent risks or cost [36]. Ambiguous and inaccurate test results can lead caretakers down the wrong path. Tests performed on apparently healthy individuals for the purpose of the early identification of underlying diseases are of variable utility, depending on the species or situation. As an example, a routine complete blood count (CBC) and biochemistry offers much insight in dolphins but that is seldom the case in tortoises. Diagnostic imaging should also be used judiciously, especially when restraint or anaesthesia is required. For example, chest and abdominal radiographs are very informative in small and medium-sized patients but in large patients they can be misleading, and they are best used as special examinations during clinical investigations in symptomatic animals rather than performed routinely on apparently healthy ones.

Manual or chemical restraint is often needed to carry out preventative medicine examinations. These inherently carry a risk of injury or death that are higher in some species than others, such as giraffes or pinnipeds. The risks are also influenced by other circumstances, such as having to dart an animal on a tall tree versus injecting it in a squeeze cage, or the technical expertise and experience of the staff. These risks need careful consideration and decisions must be justified based on evidence and consensus. The following example illustrates this: A zoo’s veterinary records reveal that, over the last two decades, two elephant seals of over eighteen years of age have succumbed to hepatic malignant neoplasia, from a total of 10 deaths during that same period. The same records reveal that the anaesthetic death rate for elephant seals of all ages is 8%. Contrast computerised tomography (CT) may allow the detection of small masses at a stage where a curative intervention may still be possible. A meeting between caretakers, veterinarians and higher management is arranged. The high risk of anaesthetic deaths, the low incidence of tumours in young animals, the real benefits of early detection and the cost of CT are discussed. Ultimately the preventative medicine program of the elephant seals is expanded with three items: a contrast computerised tomography (CT) scan under general anaesthesia every alternate year for all seals over the age of 16 years; a target for all elephant seals to voluntarily allow liver ultrasound by the age of 8 years and a review of the outcome of the changes to the elephant seals’ preventive medicine program after 3 years. The expansion of the preventive medicine is an immediate improvement of the animal welfare that is expected to present all the seals with a better opportunity to thrive and do well. External to the animals, but with the animal wellbeing in mind, using a combination of factual data and suppositions, the managers decided to improve animal welfare balancing risks, costs and expected benefits to the animals.

The considerable and ongoing advances made in the last four decades in the chemical immobilization of zoo and wildlife species have made both routine and ad hoc examinations safer for many individuals of many species. This progress has contributed significantly to improved husbandry and better veterinary practices in both the clinical and the preventative medicine fields. Preventative medicine programs require veterinary staff to spend more time with caretakers and healthy animals, multiplying the opportunities for mutual learning and positive interactions, creating deeper bonds and trust.

Table formats are well suited to scheduling the species-specific and individuals-specific requirements that must feature in the preventative medicine program. Table 2 offers an example for kinkajous, a subsection of the carnivorous mammals section of a preventative table format used by the authors.

## 3. Medical Husbandry—Integrating Operant Conditioning in Preventative Medicine Programs to Further Improve Animal Welfare

Understandably, more frequent, varied and sophisticated examinations reduce the “clinical preventable burden” [37] by increasing the chances of preventing and detecting diseases and the possibilities for early treatments [20,36]. However, some potentially useful examinations may be excluded because they would require chemical or manual restraint which present, as stated above, grave inherent risks.

Generally referred to as medical husbandry or husbandry training, the integration of operant conditioning to all aspects of husbandry and veterinary care with an emphasis on positive reinforcement is transforming the field and enhancing animal welfare and wellbeing [38,39,40,41,42,43,44,45,46,47,48,49,50,51,52,53]. By training the caretakers and the animals to perform medical behaviours, many high-risk procedures can be carried out without the need for chemical or manual restraint [53,54]. The animals become voluntary participants in their own preventative medical management [37,40,46,51,53] and willing participants in research programs [39,40,44,51,55,56], transforming stressful procedures into positive events while reducing risks and costs [38,42,43,45,46,47,48,52,53]. Medical procedures that were previously coercive and stressful are turned into rewarding and enriching experiences, increasing trust between caretakers and animals. Medical husbandry contributes to wellbeing by introducing, from the animal’s perspective, a dimension of desirability to the veterinary procedures.

Basic but valuable veterinary procedures, such as palpation, auscultation, X-rays or ultrasounds, palpation, wound-cleaning and so forth, can be performed safely and frequently. Physiological data, e.g., blood values, blood pressure, rectal temperatures, etc. can be obtained often and in a manner not affected by the method of collection, such as exertion and stress responses during capture or anaesthetic drugs.

More invasive procedures, such as injections, gastroscopy, rhinoscopy, cystoscopy [41] (Martelli, unpublished), acupuncture, intra-ocular pressure measurements, dental care, eye drops, central line catheter care, stomach tube feeding, computerised tomography (CT) scans [41] (Martelli, unpublished) and many other veterinary procedures, can be performed with the voluntary participation of the animals (Figure A1, Appendix A). Figure 2 illustrates how the inclusion of the voluntary participation of the animal increases the number and frequency of veterinary procedures while reducing risk and cost.

Improved welfare leads to better wellbeing and an increase in the duration of life in human [35,36] and in zoo populations alike [57]. As a result, geriatric medicine and the associated chronic ailments have become more prevalent [35,57] and preventative medicine is all the more relevant to the welfare and wellbeing of the animals under our care. Medical husbandry using operant conditioning improves animal welfare by reducing the risks and costs and by increasing the intensity of interventions aimed at maintaining animal health. The medical management of chronic conditions, such as diabetes, fistulae, prosthesis stumps, chronic keratitis and so forth, require frequent close-up assessment and treatments. Curative and palliative treatments of degenerative ailments can be carried out with little risk or discomfort.

Being able to include more animals in preventative medicine programs is a clear welfare gain that has allowed a deeper understanding of the origin, the prevention and the treatments of diseases. Improving animal welfare leads to long-term and prolonged wellbeing for the animals under human care. There is also evidence that experiencing positive wellbeing is conducive to seeking more positive wellbeing [32,52]. In addition, positive daily experiences lead to animals engaging in more positive interactions, such as play and collaborative behaviours, while reducing detrimental behaviours such as aggression and stereotypes [58].

## 4. Conclusions

Inevitably natural life processes, such as disease and aging, with their associated consequences, including pain, fear and death, will occur in the lives of all individuals. There are moral, ethical and legal obligations to ensure that good animal welfare practices be in place to provide the opportunities to achieve the highest possible state of wellbeing in the animals under our care.

There is a consensus that good health is an essential condition to claiming good wellbeing. Veterinary management is required to maintain or restore good health, yet veterinary preventative procedures can be invasive and can affect the animals’ comfort negatively or carry grave risks that cancel out the expected benefits.

By adopting a paradigm differentiating between animal welfare and animal wellbeing and defining animal welfare as the management systems in place to ensure maximum possible animal wellbeing, preventative medicine then emerges as an essential pillar of animal welfare, averting diseases and alerting managers of impending health problems, contributing greatly to the lasting wellbeing of the animals in the collection.

A comprehensive preventative medicine program, tailored to the zoological collection and integrated with up-to-date medical and husbandry practices, including operant conditioning, contributes immensely to animal welfare, ensuring sustained animal wellbeing. As veterinary medicine and husbandry science continue to advance, preventative veterinary procedures can be carried out more safely and more frequently. The ability to perform veterinary preventative and diagnostic procedures with the voluntary participation of the animals reduces the associated risks and costs and allows the collection of abundant and reliable physiological data, including indicators of animal wellbeing, to help objectively evaluate the adequacy of the animal welfare.

## Figures and Tables

**Figure 1 animals-13-02299-f001:**
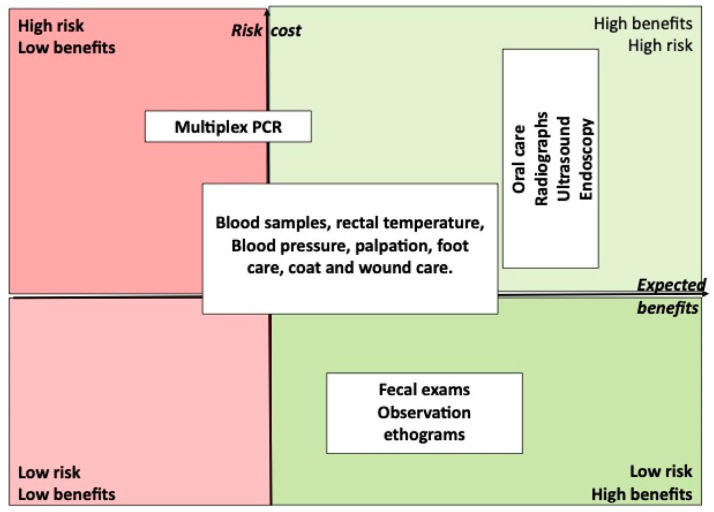
Selected procedures for the purpose of preventative medicine can be positioned within four quadrants to visualize the risk/benefit and cost benefit ratios. The technique used to carry out a procedure affects its safety and cost. The same procedure may have different benefits in different species. Therefore, the same procedure may occupy more than one quadrant. Most procedures should have a high potential benefit.

**Figure 2 animals-13-02299-f002:**
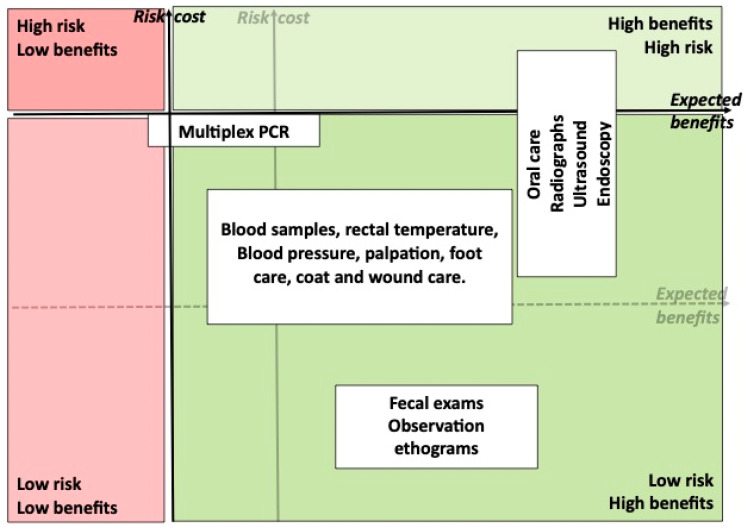
Evolution of Figure 1, the same procedures performed with the voluntary participation of the animals. The vertical risk/costs axis is shifted to the left and the horizontal expected benefit axis is shifted upwards, positioning more procedures in the high benefit–low risk quadrant.

**Table 1 animals-13-02299-t001:** Definitions of animal wellbeing and animal welfare used in this paper.

Wellbeing: The Puget Sound Institute [23] conclusions about the nature of human wellbeing can be paraphrased into a suitable working definition of animal wellbeing.	Animal wellbeing includes many aspects of an animal’s everyday life. It encompasses physical comfort, relationships with enclosure mates and caretakers and emotional and physical health. It is a state internal to the animal that is a complex individual response to many internal factors, including demographics, physical health, mental health, emotional health, social grouping dynamics, relationship with other species and staff, sex, age and external factors, such as season, weather, exhibit design, management practices.
Welfare: In this paradigm, welfare is external to the animal and is not equivalent to wellbeing nor is it a measure of the animal’s wellbeing and is analogous to human welfare [24]	Animal welfare is the sum total of the management systems in place to mitigate all inevitable negative effects of captivity and to maximize all potential benefits of human care. Animal welfare fulfils moral, ethical and legal duties of those people and organization who have control over many aspects of the lives of the animals under their care

**Table 2 animals-13-02299-t002:** Example of a working document capturing the scope and schedule of a preventative medicine program for a group of two males and one female Kinkajou (*Potos flavus*).

Prophylactic Plan 2023	Procedure	Remarks	Vet’s Comments/Follow up Action
Kinkajous: 2 males 1 female	GA for CT and general exam every 3 years	Voluntary, lying immobile in transparent box	Due in mid-2023
	Distemper, FRCPV and Rabies vaccination every 3 years.	All tolerate vaccines well; all are stable for voluntary injection	FRCPV and Rabies for all three due January 2023
	KJ23 only: suprelorin implant in June and January (4.7 mg)	Last implant in June 2022	Due in January 2023. To be confirmed, advise surgical neuter if no plan for breeding
	Heartworm prevention	Monthly oral Ivermectin	
	Faecal sample	Once a year for parasitology	Last quarter 2023
	Full body X ray under training ad hoc	Behaviour already acquired for all, to be maintained	Quarterly animal training on site with vet staff
	Abdominal ultrasound	KJ20 and KJ25 voluntary. KJ23 not fully trained	Quarterly animal training for KJ20 and KJ25, monthly for KJ23. Obtain at least one good yearly examination
	Dental visual exam under training	Voluntary, all stable to open mouth on command	January and June 2023

## Data Availability

Not applicable.

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
