# Peer review of "The Role of Preventative Medicine Programs in Animal Welfare and Wellbeing in Zoological Institutions"

_animals, 2023, doi:10.3390/ani13142299_

Round 1

Reviewer 1 Report

Overall, the utility of this study? work? (opinion piece, really) is probably high for zoo workers and less useful for others.  At least in this journal it might be seen more broadly. The thought exercise of defining and differentiating welfare from well-being, though, is valuable and a really useful way to frame these discussions in a way that is helpful to both animals and caretakers.

The simple summary and the abstract are too similar.  Are both truly required by the journal in this case?

Part 1, page 2, 4th full paragraph, there is a sentence fragment where it's unclear what message is being conveyed. (Ends in cite [32])

Page 3, first paragraph: it may be splitting hairs, but you say 3 categories, with the first 2 out of human control, but then say that humans do have an influence on the second.  Maybe state it differently in those first two sentences?

Table 1 is oddly formatted and not terribly useful.  This is probably better as another text paragraph.

Part 2, page 5, Figure 1 is a nice idea, but the myriad lines make it very confusing.  I would have the 4 quadrants, but just place boxes where you have deemed the techniques belong, rather than have so many lines.  Perhaps you could refine where these techniques reside by specifying trained vs. anesthetized? It might add more boxes, but you could more precisely place them. It is difficult to determine which is a procedure and which boxes label quadrants.

Page 5, para 2 (rolls to pg 6): this is a good argument, but not presented clearly.

Page 6, table 2, 2.1 kinkajous?  21 total animals, perhaps? There is also a little confusion in my mind about the column labels, especially the "remarks" column. I realize this might be something actually in use in your facility, but in that column, "stable" seems like a medical term applied to training.  Might there be a better way to explain this? Trained behavior? Reliable behavior?

Part 3 -- really liked addressing this means of both improving welfare and well-being. However, I felt figure 2 was unnecessary, although fun to look at.  I would also include more citations where they are relevant, especially para 3.  There are also more species than dolphins and NHPs being trained to perform health or testing related behaviors.

There are several English-related items that a competent copy editor can clear up.  For example, "Welfare" being randomly capitalized or "wellbeing" as one word.  Overall, the editors should decide yes or no for the Oxford comma. Overall, most items are clear, but there are some confusing sentences.

The OIE is now the World Organisation for Animal Health.

Page 3, 3rd and 4th paragraphs should be in concordance as to verb tense.  I would change para 4 to present tense to match para 3.

Author Response

Thank you for taking the time to review our paper. It has been resubmitted under ‘essai’.

Overall, the utility of this study? work? (opinion piece, really) is probably high for zoo workers and less useful for others.  At least in this journal it might be seen more broadly. The thought exercise of defining and differentiating welfare from well-being, though, is valuable and a really useful way to frame these discussions in a way that is helpful to both animals and caretakers.

The simple summary and the abstract are too similar.  Are both truly required by the journal in this case? We have reviewed them to make them less similar. The editors will decide if a summary is indeed necessary .

Part 1, page 2, 4th full paragraph, there is a sentence fragment where it's unclear what message is being conveyed. (Ends in cite [32]) Corrected

Page 3, first paragraph: it may be splitting hairs, but you say 3 categories, with the first 2 out of human control, but then say that humans do have an influence on the second.  Maybe state it differently in those first two sentences? Page 3 We have changed the sentence to clarify, thank you. We do indeed mean to say that humans partly influence the second category although it is not under our control. Collection managers have very little control over how each individual animal will respond mentally and physiologically to the challenges it faces but they are in charge of design, density, diets, etc

Table 1 is oddly formatted and not terribly useful.  This is probably better as another text paragraph. Table 1. We believe this table is needed but we have repositioned in text in a place that may be more appropriate. Because using a conceptual distinction between (animal) welfare and wellbeing is novel, we were advised it would be beneficial to have a table easy to refer back to.

Part 2, page 5, Figure 1 is a nice idea, but the myriad lines make it very confusing.  I would have the 4 quadrants, but just place boxes where you have deemed the techniques belong, rather than have so many lines.  Perhaps you could refine where these techniques reside by specifying trained vs. anesthetized? It might add more boxes, but you could more precisely place them. It is difficult to determine which is a procedure and which boxes label quadrants. Figures 1 and 2 have been redesigned and we believe are clearer now

Page 5, para 2 (rolls to pg 6): this is a good argument, but not presented clearly. Kindly see if the new version is satisfactory

Page 6, table 2, 2.1 kinkajous?  21 total animals, perhaps? There is also a little confusion in my mind about the column labels, especially the "remarks" column. I realize this might be something actually in use in your facility, but in that column, "stable" seems like a medical term applied to training.  Might there be a better way to explain this? Trained behavior? Reliable behavior? Thank you Table 2 needed serious work, we reformatted it slightly. Although very simple we prefer to leave it in because table formats are needed to plan and implement preventive medicine programs. Please note that x.y.z is a standard notation in zoos to describe the demographics of a group, 3.4.6 refers to a total of 13 animals, 3 males, 4 females and 6 (as yet) unknown. So 2.1 kinkajous means 2 males and 1 female. Thank you for pointing out that not all readers will be zoo people, we have changed the description

Part 3 -- really liked addressing this means of both improving welfare and well-being. However, I felt figure 2 was unnecessary, although fun to look at. Figures 1 and 2 have been redesigned and we believe are clearer now

 I would also include more citations where they are relevant, especially para 3.  There are also more species than dolphins and NHPs being trained to perform health or testing related behaviors. Your comment about the references is noted, indeed most taxa are involved in some form or other of training for health or research purpose. We did mention fish and birds (eg Kuczaj, Mattison) and have added also one very nice paper on false cobras (Williams). We intentionally refrained from having a very long bibliography, perhaps a mistake. We have 2 references in the revised text

Reviewer 2 Report

Very interesting work but quite focused in definition, ethiology of the terms...and finally the conclusion is more focused on the veterinarian role. Change the point of view of the conclusion or change the article with a different goal. 

Figure 1 Change design of the figure. It is a very interesting this kind of table but i will change squares and letters to make differences between the procedure and the title of the quadrants.

Table 2 is very specific. I suggest to create a general table with the procedures for carnivores, for example. 

There are not very connecting words during the text. i recommend to review the writing of this paper. 

Author Response

Thank you for taking the time to review our paper. It has been resubmitted under ‘essai’.

Comments and Suggestions for Authors

Very interesting work but quite focused in definition, ethiology of the terms...and finally the conclusion is more focused on the veterinarian role. Change the point of view of the conclusion or change the article with a different goal. The paper is indeed about the importance of preventative medicine for zoo animal welfare, although, as you point out there is fairly long focus on definitions. We believe that is needed because we propose to adopt a radically different paradigm that unless carefully exposed will not allow the readers to grasps how preventative programs are central to welfare.  The conclusion was revised and perhaps it may satisfy your comment about the goal of the paper.

Figure 1 Change design of the figure. It is a very interesting this kind of table but i will change squares and letters to make differences between the procedure and the title of the quadrants. Figure 1 and 2, we understand and agree with your comments, Thank you. The figures were revised to make them more clear.

Table 2 is very specific. I suggest to create a general table with the procedures for carnivores, for example. Table 2 was corrected to clarify that the example given (kinkajous) is  a subsection of the carnivores and mammals. We deliberately wanted to give an example of a utilitarian table because trying to recommend a prescriptive preventative medicine for all carnivores is outside of the scope of this paper and also somewhat in contradiction with the statements that such programs are best tailored to the ‘parochial’ needs of individual institutions

Comments on the Quality of English Language. There are not very connecting words during the text. i recommend to review the writing of this paper. Some rewriting was carried out, hopefully improving overall readability

Reviewer 3 Report

In this manuscript, the authors have extensively reviewed the primary objective of a preventative medicine program. This is to minimize the likelihood of health issues arising and maximize the chances of early detection, with the ultimate aim of benefiting both the animals and the organization. First, they expose conventional concepts of welfare and wellbeing, and they state that it becomes necessary to establish a paradigm that distinguishes animal well-being from animal welfare, enabling clear thinking and guiding actions concerning the welfare of animals in human care. By embracing such a model, preventative medicine programs emerge as a fundamental aspect of promoting the welfare of zoo and aquarium animals, particularly when they incorporate modern veterinary and husbandry techniques, including operant conditioning.

I find the subject matter intriguing because the manuscript highlights a distinction between animal welfare and animal well-being. This differentiation has caught my attention and piqued my curiosity. The discussion surrounding the role of preventative medicine programs in promoting animal welfare is an important and complex topic. There are several points to consider when examining this issue.

Firstly, the overarching goal of a preventative medicine program is indeed to minimize the chances of health problems developing and to detect health problems early. This is crucial for ensuring the wellbeing of animals under human care, as early detection and intervention can prevent suffering and improve outcomes.

However, as the authors state, traditional paradigms of animal welfare may not perfectly align with animals undergoing veterinary treatments, including quarantine and preventative medicine. When animals receive veterinary care, their physiology and behavior may be altered from their normal state, and they may experience limitations in their choices and comfort. This raises questions about how we define and assess animal welfare in these specific contexts. I would like the authors to delve deeper into this topic and explore relevant literature to establish some guidelines for managing animals during these critical moments in order to minimize the negative consequences of such veterinary interventions.

I also find it necessary to separate animal wellbeing from animal welfare. This approach emphasizes the need for a clear understanding of the physical and psychological wellbeing of animals, while acknowledging that their welfare may be compromised during certain veterinary procedures. By adopting this paradigm, we can strive to ensure that animals receive the necessary care and treatments while also considering their overall wellbeing.

As indicated here, incorporating modern veterinary and husbandry techniques, such as operant conditioning, can play a significant role in promoting animal welfare within preventative medicine programs. These techniques allow for positive reinforcement, training, and behavioral management, which can enhance the animals' experience and minimize stress during veterinary procedures. The authors should cite relevant literature on positive reinforcement operant conditioning to guide readers in implementing these training techniques within captive zoo collections.

Overall, the discussion revolves around finding a balance between the necessary medical interventions and the wellbeing of animals. It requires careful consideration of the specific needs and circumstances of each animal, as well as a commitment to continually improve practices and approaches to ensure the highest standards of animal welfare within preventative medicine programs.

This manuscript is well structured and synthesized; however, I don't believe this to be a research article as there is no original research presented in the manuscript. Instead, it could be categorized as a review or commentary publication.

On the other hand, some minor issues need to be addressed:  

- Keywords: avoid using same words as in the title.

- Some format issues must be corrected.

- Please, check double spacing throughout the entire text

- Please do not capitalize “Welfare” or “Animal” where it is not necessary (page 2, page 3…)

- page 2: In the text: "definition of welfare does not distinguish between welfare and wellbeing either when it states that welfare is is “how an animal copes with the conditions in which it lives”". Delete one word “is”.

Author Response

Thank you for taking the time to review this paper.  The revision will be resubmitted under "essai" instead of 'article'.

Comments and Suggestions for Authors

In this manuscript, the authors have extensively reviewed the primary objective of a preventative medicine program. This is to minimize the likelihood of health issues arising and maximize the chances of early detection, with the ultimate aim of benefiting both the animals and the organization. First, they expose conventional concepts of welfare and wellbeing, and they state that it becomes necessary to establish a paradigm that distinguishes animal well-being from animal welfare, enabling clear thinking and guiding actions concerning the welfare of animals in human care. By embracing such a model, preventative medicine programs emerge as a fundamental aspect of promoting the welfare of zoo and aquarium animals, particularly when they incorporate modern veterinary and husbandry techniques, including operant conditioning.

I find the subject matter intriguing because the manuscript highlights a distinction between animal welfare and animal well-being. This differentiation has caught my attention and piqued my curiosity. The discussion surrounding the role of preventative medicine programs in promoting animal welfare is an important and complex topic. There are several points to consider when examining this issue.

Firstly, the overarching goal of a preventative medicine program is indeed to minimize the chances of health problems developing and to detect health problems early. This is crucial for ensuring the wellbeing of animals under human care, as early detection and intervention can prevent suffering and improve outcomes.

However, as the authors state, traditional paradigms of animal welfare may not perfectly align with animals undergoing veterinary treatments, including quarantine and preventative medicine. When animals receive veterinary care, their physiology and behavior may be altered from their normal state, and they may experience limitations in their choices and comfort. This raises questions about how we define and assess animal welfare in these specific contexts. I would like the authors to delve deeper into this topic and explore relevant literature to establish some guidelines for managing animals during these critical moments in order to minimize the negative consequences of such veterinary interventions. In regards expanding on minimising degraded wellbeing due to preventative veterinary procedures the two take home messages made in the paper are that welfare programs carefully consider the ratio of risks over expected yield  and that operant conditioning be incorporated to change procedures from imposed stressors into voluntary rewarding events. The matter of delving further into how to define welfare under the unpleasant conditions that accompany disease and veterinary procedures is what prompted us to adopt a paradigm where there is a strict distinction between welfare and wellbeing. The welfare aspect of veterinary preventative programs is in the soundness and value of the programs to the overall clinical burden avoided and resources spent.

I also find it necessary to separate animal wellbeing from animal welfare. This approach emphasizes the need for a clear understanding of the physical and psychological wellbeing of animals, while acknowledging that their welfare may be compromised during certain veterinary procedures. By adopting this paradigm, we can strive to ensure that animals receive the necessary care and treatments while also considering their overall wellbeing.

As indicated here, incorporating modern veterinary and husbandry techniques, such as operant conditioning, can play a significant role in promoting animal welfare within preventative medicine programs. These techniques allow for positive reinforcement, training, and behavioral management, which can enhance the animals' experience and minimize stress during veterinary procedures. The authors should cite relevant literature on positive reinforcement operant conditioning to guide readers in implementing these training techniques within captive zoo collections. Besides the original reference specific to operant conditioning techniques (Ramirez) we have added a reference on operant conditioning that also encourage the reader to implement it (Lukas)

Overall, the discussion revolves around finding a balance between the necessary medical interventions and the wellbeing of animals. It requires careful consideration of the specific needs and circumstances of each animal, as well as a commitment to continually improve practices and approaches to ensure the highest standards of animal welfare within preventative medicine programs.

This manuscript is well structured and synthesized; however, I don't believe this to be a research article as there is no original research presented in the manuscript. Instead, it could be categorized as a review or commentary publication. Agreed it will be submitted under Essai

On the other hand, some minor issues need to be addressed:  

- Keywords: avoid using same words as in the title.

- Some format issues must be corrected.

- Please, check double spacing throughout the entire text

- Please do not capitalize “Welfare” or “Animal” where it is not necessary (page 2, page 3…)

- page 2: In the text: "definition of welfare does not distinguish between welfare and wellbeing either when it states that welfare is is “how an animal copes with the conditions in which it lives”". Delete one word “is”.

We have tried improving on the English language and checked capitalization and other items identified by the reviewers, thank you
